

# Highly pathogenic avian influenza virus of the A/H5N8 subtype, clade 2.3.4.4b, caused outbreaks in Kazakhstan in 2020

Asylulan Amirgazin[1], Alexandr Shevtsov[1], Talgat Karibayev[2], Maxat Berdikulov[2], Tamila Kozhakhmetova[2], Laura Syzdykova[1], Yerlan Ramankulov[1,3] and Alexandr V. Shustov[1]

[1] National Center for Biotechnology, Nur-Sultan, Akmola Region, Kazakhstan
[2] National Reference Veterinary Center, Nur-Sultan, Akmola Region, Kazakhstan
[3] National Laboratory Astana, Nazarbayev University, Nur-Sultan, Akmola Region, Kazakhstan

## ABSTRACT

**Background**. Large poultry die-offs happened in Kazakhstan during autumn of 2020. The birds' disease appeared to be avian influenza. Northern Kazakhstan was hit first and then the disease propagated across the country affecting eleven provinces. This study reports the results of full-genome sequencing of viruses collected during the outbreaks and investigation of their relationship to avian influenza virus isolates in the contemporary circulation in Eurasia.

**Methods**. Samples were collected from diseased birds during the 2020 outbreaks in Kazakhstan. Initial virus detection and subtyping was done using RT-PCR. Ten samples collected during expeditions to Northern and Southern Kazakhstan were used for full-genome sequencing of avian influenza viruses. Phylogenetic analysis was used to compare viruses from Kazakhstan to viral isolates from other world regions.

**Results**. Phylogenetic trees for hemagglutinin and neuraminidase show that viruses from Kazakhstan belong to the A/H5N8 subtype and to the hemagglutinin H5 clade 2.3.4.4b. Deduced hemagglutinin amino acid sequences in all Kazakhstan's viruses in this study contain the polybasic cleavage site (KRRKR-G) indicative of the highly pathogenic phenotype. Building phylogenetic trees with the Bayesian phylogenetics results in higher statistical support for clusters than using distance methods. The Kazakhstan's viruses cluster with isolates from Southern Russia, the Russian Caucasus, the Ural region, and southwestern Siberia. Other closely related prototypes are from Eastern Europe. The Central Asia Migratory Flyway passes over Kazakhstan and birds have intermediate stops in Northern Kazakhstan. It is postulated that the A/H5N8 subtype was introduced with migrating birds.

**Conclusion**. The findings confirm the introduction of the highly pathogenic avian influenza viruses of the A/Goose/Guangdong/96 (Gs/GD) H5 lineage in Kazakhstan. This virus poses a tangible threat to public health. Considering the results of this study, it looks justifiable to undertake measures in preparation, such as install sentinel surveillance for human cases of avian influenza in the largest pulmonary units, develop a human A/H5N8 vaccine and human diagnostics capable of HPAI discrimination.

Corresponding author
Alexandr V. Shustov,
shustov@biocenter.kz

## INTRODUCTION

Influenza viruses of type A are members in the genus Alphainfluenzavirus in the Orthomyxoviridae family. Viruses in this species that can infect and productively circulate in birds are called avian influenza viruses (AIVs). Naturally, aquatic birds and shorebirds, and more specifically representatives from avian orders Anseriformes and Charadriiformes, are the natural reservoir for AIVs (*Olsen et al., 2006*; *Venkatesh et al., 2018*).

AIVs can be highly contagious for gallinaceous birds including poultry (chicken, turkey). Variants dubbed highly pathogenic avian influenza viruses (HPAI) cause severe disease in gallinaceous birds sometimes resulting in massive birds' die-offs and devastating economic consequences for poultry farms (*Alexander, 2000*). Cross-species transmission from birds to mammals or humans is infrequent, however human infections with AIVs may be severe, with high case-fatality rate (*Peiris, De Jong & Guan, 2007*).

The AIVs occasionally participate in the reassortment of genome segments. Reassortment between AIVs circulating in the same host-species is a recognized driver for the genetic diversity and evolution (*Postnikova et al., 2021*). On the contrary, reassortment between viruses from different host classes, such as avian and mammalian viruses, is uncommon (*Ganti et al., 2021*; *Schrauwen et al., 2013*). Importantly, rare events when a human virus acquired one or more gene segments from AIV resulted in pandemics (*Schrauwen & Fouchier, 2014*; *Webster et al., 1992*). Surveillance systems across the world share pervasive interest in the AIVs circulation and evolution.

Surface proteins hemagglutinin (HA) and neuraminidase (NA) determine the antigenic properties of the virus. AIVs are divided into subtypes determined by the serological reactivity of hemagglutinin (18 HA subtypes) and neuraminidase (11 NA subtypes) (*Kosik & Yewdell, 2019*).

Some AIVs cause rare but severe and life-threatening infections in humans, involving respiratory, gastrointestinal and neural systems (*Wong & Yuen, 2006*). A limited number of AIV subtypes seem to be capable of infecting humans, because until recently only certain combinations of avian HA (H5, H6, H7, H9 and H10) with NA (N1, N2, N3, N4, N6, N7, N8, N9), but excluding A/H5N8, have been reported in humans (*Mostafa et al., 2018*; *Fouchier et al., 2004*; *Lin et al., 2000*; *Munster et al., 2007*; *Peiris et al., 1999*; *Abdel-Ghafar et al., 2008*). However, the subtype A/H5N8 was isolated from humans contacting poultry in 2020 (*Pyankova et al., 2021*). A number of registered human A/H5N8 cases in that outbreak totaled to seven (*World Health Organization, 2021*).

In Republic of Kazakhstan, starting from autumn 2020 there was a series of large outbreaks of a bird's disease that appeared to be avian influenza (*Sputnik, 2021*). The first registered outbreaks occurred in settlements along the Kazakhstan-Russia border. To the year-end, the outbreaks have been registered in eleven country's provinces ('oblast').
Among them, provinces of Northern Kazakhstan were the most disease-affected regions. The state surveillance system for avian influenza in Kazakhstan monitored the outbreaks. The veterinary service made efforts to slow down disease propagation, set quarantines in affected settlements, imposed restrictions on the export of poultry and poultry products, organized the prompt vaccination of remaining poultry. In 2021, poultry outbreaks ceased, and as of the time this manuscript has been written the country is declared AIV-free.

The goal of this study was to identify the closest relatives for the viruses that caused outbreaks in Kazakhstan among contemporary isolates from other world regions, to confirm the existence of the common circulation and postulate source(s) of introduction. Other goal was to describe mutations in Kazakhstan's variants as compared to prototype isolates.

## MATERIALS & METHODS

### Collecting samples for the study

The main reference laboratory in the Kazakhstan's state system for veterinary control and surveillance, the National Reference Veterinary Center (NRVC), organized expeditions to provinces in Northern and Southern Kazakhstan, where outbreaks of avian influenza were officially registered. The NRVC veterinarians monitored areas within a 10 km-radius from poultry farms or disease-affected private households, as well as around settlements officially designated as "disease reservoirs". Birds' carcasses or visibly diseased birds were examined. In total, 2212 birds were examined and used for samples collection. Samples were brains, lungs, trachea, spleen, intestinal tract and feces. The vast majority of the samples were from chickens, domestic ducks and geese. Other samples were from mute swan (*Cygnus olor*), gray crow (*Corvus cornix*), dove (*Columba livia*). Veterinarians diagnosed the disease in live birds based on physical signs such as coma or prostration, unnatural position of head and neck, difficulty breathing, refusal to feed, hyperemia and swelling of visible mucous membranes, flows from beak and nasal openings, diarrhea. When examining carcasses, attention was paid to hemorrhages in the mucous membranes, under serous membranes of internal organs, in the subcutaneous tissue or in the muscles, the presence of enteritis, conjunctivitis or inflammation of other internal organs, pulmonary edema.

### Molecular diagnostics and virus subtyping

Frozen samples were transported to NRVC where PCR-diagnostics was performed. Pieces of organs were homogenized using TissueLyser bead mill (Qiagen, Hilden, Germany) to produce 10% suspensions in saline (0.9% NaCl). The suspensions were transferred in 1.5 ml tubes and centrifuged at 10,000 rpm for 30 s. The supernatants were used for RNA extraction. Samples of feces were suspended to obtain 10% suspensions in saline. The suspensions were centrifuged and the supernatants were used for RNA extraction. Total RNA was purified using QIAamp Viral RNA Kits (Cat# 52906, Qiagen, Germany) as per the manufacturer's instructions.

Viral RNA was detected using a test-system for real-time PCR (RT-PCR), namely PCR-INFLUENZA-A-FACTOR (Cat# R10515-VET, LLC "VET FACTOR", Troitsk, Russia). This test-system targets for amplification conserved (subtype-independent) sequences in

the MP segment in the viral genome. Another kit, PCR-FLU-TYPE-H5/H7/H9-FACTOR (Cat# R12817-VET, LLC "VET FACTOR", Troitsk, Russia) was used for HA typing. The kit allows assigning HA subtypes H5, H7, H9 based on results of one multiplexed reaction. Thermal cycler Rotor Gene 6000 was used to conduct the assay. Neuraminidase (NA) subtype was determined using sequencing and identification of the closest prototypes.

## Whole genome sequencing

Ten samples collected by NRVC veterinarians during their visits to the most disease-affected settlements were selected for sequencing the full AIV genome. Larger utilization of the full-genome sequencing was not possible due to the high costs. The samples were transferred to the National Center for Biotechnology (NCB). Reverse transcription was performed using SuperScript IV Reverse Transcriptase (Cat# 18090010, Invitrogen, CA). The first strand-cDNA was primed using a universal primer MBTuni-12 (5′-ACGCGTGATCAGCAAAAGCAGG-3′). Eight segments in the AIV genome were PCR-amplified using a pair of MBTuni-12 and MBTuni-13 (5′-ACGCGTGATCAGTAGAAACAAGG-3′) (*Zhou et al., 2009*). These primers target conserved sequences at the 5′- and 3′-termini of genomic segments. The high-fidelity DNA polymerase Phusion (Cat# F-530XL, Invitrogen, CA) was used in PCR. Amplification products were purified using AMPure magnetic particles (Cat# A63880, Beckman Coulter, USA) according to the manufacturer's instructions.

A fragment library was prepared using Nextera DNA Flex Library Prep Kit (20015829, Illumina, USA). Then, the MiSeq Reagent Kit v3, 600 cycles (MS-102-3003, Illumina, USA) and MiSeq platform (Illumina) were used to conduct the sequencing.

In addition to ten samples sequenced by the authors of this work, six different samples from the year-2020-outbreaks were sequenced in foreign laboratories, namely the Animal and Plant Health Agency (APHA) (Addlestone, United Kingdom) and WHO National Influenza Centre (WHO NIC) in St. Petersburg (Russia). In order to increase the representation of viruses from Kazakhstan in the study, we downloaded the APHA- and WHO NIC-generated sequences from GISAID (*GISAID, 2022*; *Elbe & Buckland-Merrett, 2017*; *Shu & McCauley, 2017*) and included in the analysis in this work. The contribution from the originating laboratories is acknowledged in the Supplementary Information.

## Bioinformatics and phylogenetic analysis

Quality of sequencing reads was assessed using the FastQC v0.11.9 program (*Babraham Institute, 2019*). Trimming to quality values Q30 was performed using the SeqTK v1.3-r117/Sickle suite (*Shen et al., 2016*). The sequencing reads were assembled into contigs using the BWA tool (*Li & Durbin, 2010*). The following reference sequences were used as guides during the assembly of contigs: isolate A/White-fronted Goose/AN/1-15-12/2016 (Genbank entries: MH988775.1 for HA, MH989503.1—NA); and isolate A/chicken/Omsk/0112/2020 (GISAID accessions: EPI1813342—PB2, EPI1813343—PB1, EPI1813341—PA, EPI1813338—NP, EPI1813340—MP, EPI1813339—NS). Consensus sequences were produced using FreeBayes (*FreeBayes, 2021*) and BCFtools (*Danecek et al., 2021*). The complete genomes of Kazakhstan's viruses determined in our laboratory were deposited in GISAID. Our entries are listed in the Supplementary Information.

Sequences of historic prototypes and isolates in contemporary circulation were downloaded from GISAID, their listing with acknowledgements to originating laboratories is provided in the Supplementary Information. In the presented phylogenetic trees, viruses are identified by the GISAID isolate name (EPI_ISL) and accession numbers (EPI#) for particular segments. The goal of this study was to identify isolates in the global circulation which share the closest similarity to viruses from the recent outbreaks in Kazakhstan, and hence, to point to a possible path of introduction. Accordingly, all available complete-genome sequences from Russia, Europe, China and Asia-Pacific collected between Jan. 2020–Aug. 2021 (*i.e.,* in the time frame overlapping the Kazakhstan's outbreaks) were used in the initial analysis. For Africa and Middle East, sequences deposited from 2018 to 2021 were used because of the shortage of recent (2020-2021) submissions from these regions. The phylogenetic trees were constructed using the Bayesian probabilistic inference with Markov Chain Monte Carlo (MCMC) sampling algorithms in MrBayes v.3.2.7a (*Ronquist et al., 2012*). The initial analysis was performed with a number of evolutionary process models, *i.e.,* the Jukes-Cantor (JC69) model, generalized time-reversible (GTR) model, and GTR with rate variation across sites (GTR+G) with four rate categories. The initial analysis was done using chain length of 1,000,000. Models showing good convergence such as an estimated sample size (ESS) of at least 1000, potential scale reduction factor (PSRF) close to one (PSRF = 1), and also computation times <24 h, were used in the further analysis. Phylogenetic trees and support statistics for the branches presented in the paper and Supplementary Information were constructed using the MCMC chain length of 10,000,000. With the final settings, ESS values were universally above 10,000 and PSRF = 1 indicating perfect convergence. Consensus trees (50% majority rule trees) are presented in the paper and the Supplementary Information. Posterior probability percents are shown near the nodes to indicate the statistical support for clustering. The H5 HA clades naming and labeling HA tree-branches as the clades 2.3.4.4a–2.3.4.4 h are in accordance to the recent nomenclature (*Anonymus, 2022*).

FluSurver (*A\*STAR Bioinformatics Institute, 2022*) was used to identify mutations in the influenza virus surface proteins which possibly influence biological characteristics such as host specificity, drugs resistance, virulence, etc.

## RESULTS

### Bird flu in Northern Kazakhstan, 2020

During autumn 2020, unusually high numbers of wild birds' carcasses and also live but dying birds were noticed in many places in provinces of Northern Kazakhstan. Local veterinarians reported that initial die-offs started in wild wetland birds which had been migrating southwards. However, exact losses to wild birds' populations are unknown because no attempts were made to quantify diseased or dead birds in wild habitats. Soon, die-offs in poultry began and attracted attention of national media and officials. Affected poultry were chickens, ducks, geese and turkeys. First officially registered outbreaks in poultry are dated in September 2020. The situation with the disease deteriorated during autumn of 2020, because the disease propagated across the country and increasing numbers
of farms faced poultry die-offs. According to data in the official NRVC report, the headcount of diseased birds in private households and commercial farms totaled 69,828 (National Reference Veterinary Center, 2021, unpublished report: http://www.nrcv.kz/). Destruction of all poultry in affected farms was mandatory. The total loss of poultry was 2 million at the end of 2020 including the culling (*Sputnik, 2021*). Many farms stopped production for about 3 months resulting in an increase in market prices for poultry products (*EurasiaNet, 2020*). Eleven out of fourteen Kazakhstan provinces had reported outbreaks before the country was declared avian flu-free. Four provinces of Northern Kazakhstan (Akmola, Kostanay, Pavlodar, North-Kazakhstan) had the largest numbers of outbreaks.

The state surveillance system for avian influenza collected samples from 2212 bird's carcasses. All the samples were tested for AIV using a commercial RT-PCR test system. Total number of PCR-positives amounted to 1976. Ten samples were taken from bird carcasses with overt pathological signs compatible with avian influenza. The samples were collected by NRVC specialists during on-site visits during different expeditions to North Kazakhstan and the Almaty province (Southern Kazakhstan). Map in Fig. 1 shows sampling locations. The ten samples were utilized for full-genome sequencing of Kazakhstan's AIVs.

## Molecular variability of surface glycoproteins

Surface glycoproteins determine antigenic properties of the virus and undergo driving selection under the pressure of the host immune response. Comparing the HA nucleotide sequences within the group of Kazakhstan's viruses identifies 32 variable positions (per 1704 sequenced positions). Similar comparison of the NA nucleotide sequences shows 36 polymorphic positions (per 1413 nt length). The majority of the nucleotide mutations are silent, as only four polymorphic sites were found at the amino acid level in HA, and 12 polymorphic sites were found in deduced NA amino acid sequences. Data on the amino acid mutations found in surface glycoproteins of Kazakhstan's viruses are presented in Tables 1–2 for HA, and Tables 3–4 for NA.

The deduced HA amino acid sequences contain the same cluster of basic amino acid residues (KRRKR-G) in all Kazakhstan viruses. This is the cleavage site separating two HA subunits (HA1 and HA2). The presence of such "polybasic cleavage site" is considered to be a major virulence determinant and associated with the HPAI phenotype (*Tscherne & García-Sastre, 2011*).

The FluSurver service developed by the A\*STAR Bioinformatics Institute in Singapore is a tool allowing identifying and annotating mutations in influenza virus proteins which may change virus' properties such as host range, drug resistance, etc. As compared to the nearest (FluSurver-proposed) reference strain A/Sichuan/26221/2014(H5N6), the HA glycoproteins in Kazakhstan's viruses share common mutations: K3N, G16S, N110S, T139P, T156A, Q185R, V194I, A201E, N252D, E284G, M285V, I298V, K492E, V538A, I547M and V548M. In addition to the common mutations, two viruses (A/goose/Kazakhstan/4-190-20-B-H5N8-1/2020 and A/chicken/Kazakhstan/220-B-2-H5N8-4/2020) have S12G.

The NA sequences were screened for the presence of biologically relevant mutations using the service FluSurver. As compared to the nearest service-identified prototype A/Baikal

![PeerJ]

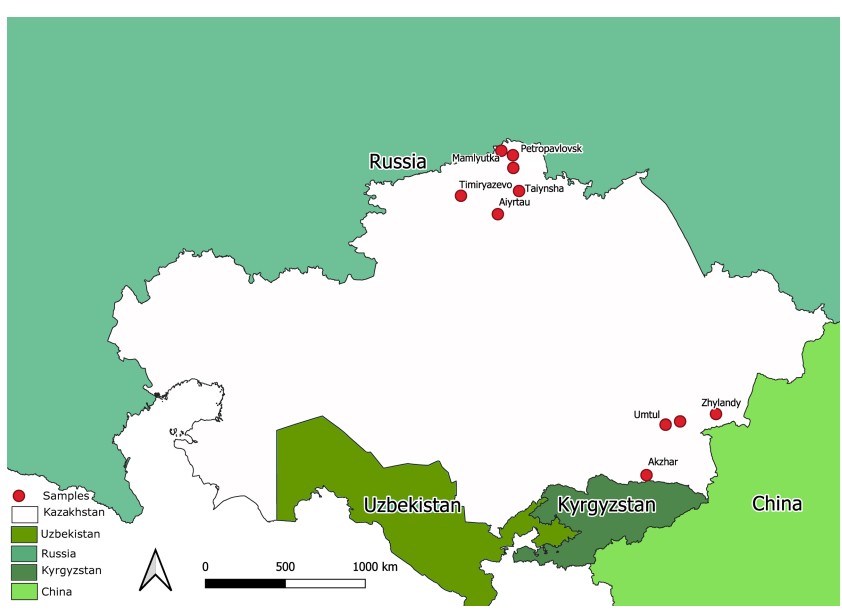

**Figure 1** Map of Kazakhstan (with adjacent territories of neighbor countries) showing locations where samples were collected for full-genome sequencing.

**Table 1 Occurrence of amino acid mutations in hemagglutinin in AIVs from Kazakhstan.**

| Amino acid position | Polymorphism | Allelic frequencies (%) | |
|---|---|---|---|
| 12 | S/G | S (87.5) | G (12.5) |
| 88 | R/K | R (93.8) | K (6.3) |
| 166 | L/I | L (93.8) | I (6.3) |
| 205 | N/T | N (93.8) | T (6.3) |

teal/Korea Donglim/3/2014(H5N8), Kazakhstan's viruses share common mutations V8A, V27L, T32M, N46K, V49I, T81A, V106I, S136A, A190T, V201I, V213I, A245S, G263D, R264Q, T265A, T295M, I303V, T329A, V359M, S397L, Y450H, and K469G. Mutations V71I, E72K, N84S, P88S, Q330R, and K469E are present in a small fraction (10–20%) of our collection. Additionally, 40% of our viruses have K141R.

## Relation of Kazakhstan's viruses to isolates in contemporary circulation

The main goal of phylogenetic analysis in this work was to identify isolates in the contemporary circulation which are the closest relatives to Kazakhstan's viruses. In preliminary experiments we compared different approaches for building phylogenetic trees and calculating the statistical support for branches. The Maximum Likelihood (ML) method with the Bayesian probabilistic selection of the best model-fitting phylogenies, produced the trees with more robust clustering as compared to simple distance-based methods (Minimum Evolution (ME), etc.). Statistical support values for branches in Bayesian-MCMC-phylogenetic trees are the posterior probabilities, and these universally

**Table 2  List of amino acid mutations found in hemagglutinin in AIVs from Kazakhstan.**

| Name, Accession | Position | | | |
|---|---|---|---|---|
| | 12 | 88 | 166 | 205 |
| A/goose/Kazakhstan/4-190-20-B-H5N8-1/2020 EPI1882525 | G | R | L | N |
| A/chicken/Kazakhstan/220-B-2-H5N8-4/2020 EPI1882547 | G | R | L | N |
| A/duck/Kazakhstan/12-20-B-Talg-11/2020 EPI1882548 | S | R | L | N |
| A/goose/Kazakhstan/7-20-B-Talg-12/2020 EPI1882551 | S | R | L | N |
| A/swan/Kazakhstan/9-20-B-Talg-39/2020 EPI1882552 | S | R | L | N |
| A/chicken/Kazakhstan/12-20-B-Talg-45/2020 EPI1882555 | S | R | L | N |
| A/crow/Kazakhstan/15-20-B-Talg-4/2020 EPI1882556 | S | R | L | N |
| A/swan/Kazakhstan/1-267-20-B-Talg-52/2020 EPI1882559 | S | R | L | N |
| A/pigeon/Kazakhstan/15-20-B-Talg-5/2020 EPI1882560 | S | R | L | N |
| A/chicken/Kazakhstan/1-20-B-Talg-67/2020 EPI1882563 | S | R | L | N |
| A/domestic goose/Kazakhstan/1-248_2-20-B/2020 EPI1811601 | S | R | I | T |
| A/chicken/Kazakhstan/Kn-3/2020 EPI1839261 | S | R | L | N |
| A/chicken/Kazakhstan/Kn-6/2020 EPI1839269 | S | R | L | N |
| A/domestic duck/Kazakhstan/1-274-20-B/2020 EPI1811611 | S | K | L | N |
| A/domestic goose/Kazakhstan/1-242_2-20-B/2020 EPI1811619 | S | R | L | N |
| A/mute swan/Kazakhstan/1-267-20-B/2020 EPI1811584 | S | R | L | N |

**Table 3  Occurrence of amino acid mutations in neuraminidase in AIVs from Kazakhstan.**

| Amino acid position | Polymorphism | Allelic frequencies (%) | |
|---|---|---|---|
| 9 | T/A | T (93.8) | A (6.3) |
| 70 | V/I | V (93.8) | I (6.3) |
| 71 | V/I | V (87.5) | I (12.5) |
| 72 | E/K | E (93.8) | K (6.3) |
| 84 | N/S | N (93.8) | S (6.3) |
| 88 | P/S | P (93.8) | S (6.3) |
| 141 | K/R | K (75.0) | R (25.0) |
| 175 | V/L | V (93.8) | L (6.3) |
| 330 | Q/R | Q (87.5) | R (12.5) |
| 449 | D/E | D (87.5) | E (12.5) |
| 466 | D/N | D (93.8) | N (6.3) |
| 469 | G/E | G (87.5) | E (12.5) |

have been higher than bootstrap values commonly used with distance methods. This justifies the use of the Bayesian MCMC approach. The trees presented in Figs. 2 and 3 and figures in the Supplementary Information have been produced using the Bayesian MCMC modeling.

Phylogenetic trees for the segments HA and NA show that all sixteen viruses from Kazakhstan belong to the A/H5N8 subtype. The HA H5 clade 2.3.4.4b comprises all viruses from Kazakhstan (Fig. 2). The H5 clade 2.3.4.4b had evolved at around 2016 and currently this clade predominates worldwide (*World Health Organization, 2018*; *Lee et al., 2017b*; *Lee*

**Table 4  List of amino acid mutations found in neuraminidase in AIVs from Kazakhstan.**

| Name, Accession | Position | | | | | | | | | | | |
|---|---|---|---|---|---|---|---|---|---|---|---|---|
| | 9 | 70 | 71 | 72 | 84 | 88 | 141 | 175 | 330 | 449 | 466 | 469 |
| A/goose/Kazakhstan/4-190-20-B-H5N8-1/2020 EPI1882526 | T | V | V | E | S | P | K | V | Q | D | D | G |
| A/chicken/Kazakhstan/220-B-2-H5N8-4/2020 EPI1882546 | T | V | V | E | S | P | K | V | Q | D | D | G |
| A/duck/Kazakhstan/12-20-B-Talg-11/2020 EPI1882549 | T | V | V | E | N | P | R | V | Q | D | D | G |
| A/goose/Kazakhstan/7-20-B-Talg-12/2020 EPI1882550 | T | V | V | E | N | P | R | V | Q | D | D | G |
| A/swan/Kazakhstan/9-20-B-Talg-39/2020 EPI1882553 | T | V | V | K | N | P | K | V | Q | D | D | G |
| A/chicken/Kazakhstan/12-20-B-Talg-45/2020 EPI1882554 | T | V | V | E | N | S | K | V | Q | D | D | G |
| A/crow/Kazakhstan/15-20-B-Talg-4/2020 EPI1882557 | T | V | V | E | N | P | R | V | Q | D | D | G |
| A/swan/Kazakhstan/1-267-20-B-Talg-52/2020 EPI1882558 | T | V | I | E | N | P | K | V | R | D | D | E |
| A/pigeon/Kazakhstan/15-20-B-Talg-5/2020 EPI1882561 | T | V | V | E | N | P | R | V | Q | D | D | G |
| A/chicken/Kazakhstan/1-20-B-Talg-67/2020 EPI1882562 | T | V | V | E | N | P | K | V | Q | D | D | G |
| A/domestic goose/Kazakhstan/1-248_2-20-B/2020 EPI1811603 | A | V | V | E | N | P | K | L | Q | D | D | G |
| A/chicken/Kazakhstan/Kn-3/2020 EPI1839260 | T | V | V | E | N | P | K | V | Q | E | D | G |
| A/chicken/Kazakhstan/Kn-6/2020 EPI1839268 | T | V | V | E | N | P | K | V | Q | E | D | G |
| A/domestic duck/Kazakhstan/1-274-20-B/2020 EPI1811613 | T | V | V | E | N | P | K | V | Q | D | D | G |
| A/domestic goose/Kazakhstan/1-242_2-20-B/2020 EPI1811621 | T | V | I | E | N | P | K | V | R | D | D | E |
| A/mute swan/Kazakhstan/1-267-20-B/2020 EPI1811586 | T | I | V | E | N | P | K | V | Q | D | N | G |

*et al., 2017a*; *Chen et al., 2019*; *Le & Nguyen, 2014*; *Yamaji et al., 2020*; *Gu et al., 2013*; *Tarek et al., 2021*; *Zhang et al., 2020*; *Pohlmann et al., 2019*; *Laleye et al., 2021*; *Lewis et al., 2021*; *Globig et al., 2017*; *Liang et al., 2020*; *Baek et al., 2021*).

In phylogenetic trees (Figs. 2 and 3 and the Supplementary Information) Kazakhstan's viruses cluster with isolates from Southern Russia (*e.g.*, Rostov-on-Don, Krasnodar), Russian Caucasus (North Ossetia-Alania), Ural region (Chelyabinsk), South-Western Siberia (Omsk). Other closely related prototypes are from Eastern Europe.

In the HA tree, the representatives from Kazakhstan group into five clusters with high statistical support. These clusters are provisionally labeled as branches 1–5 (B1–B5) in Fig. 2. The given labeling is used hereafter to describe the findings. The posterior probability percents for branches B1–B5 are quite high (99–100%). Also, the branch B5 may be divided into lower-order clusters and a single-tip branch. Some of the lower-order branches (labeled B5-1, B5-2, B5-3 in Fig. 2) also have high statistical support (100%). With this regard, the clusterization of the NA sequences follows a very similar pattern. More specifically, branches B1–B4 in the NA tree (Fig. 3) have high statistical support. The latter comprise the same representatives from Kazakhstan as the eponymous branches in the HA tree. Sequences from the cluster B5 in the HA tree appear not to form a separate cluster in the NA tree. However, all studied NA sequences from Kazakhstan do form a mixed group with sequences from South-Western Siberia and Eastern Europe, and this group has significant statistical support (73%).

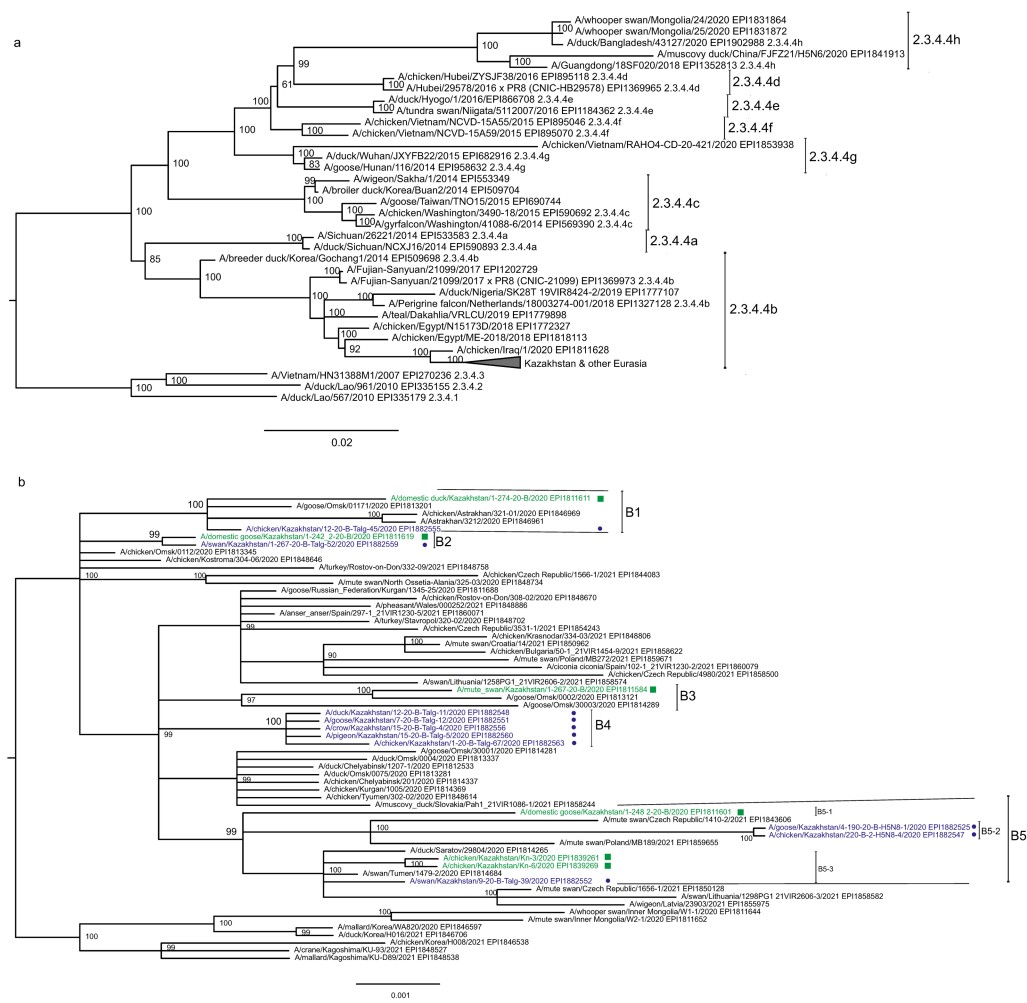

**Figure 2   Phylogenetic tree for the hemagglutinin (HA) gene.** The tree is split in two parts, each part in a separate panel. The consensus tree was built using the Bayesian inference as described in the Materials and Methods. Branch tips are labeled with the GISAID isolate name (EPI_ISL), accession number (EPI) and additional markings. Numbers at tree nodes show the statistical support (posterior probability percent). A multiple alignment used to build this tree is provided in the Supplementary Information. (A) HPAI A/H5N8 viruses from Kazakhstan year-2020 outbreaks cluster together with contemporary isolates from Eurasia and belong to the H5 clade 2.3.4.4b. Hemagglutinin clades 2.3.4.4a–2.3.4.4 h as well as clades 2.3.4.1–2.3.4.3 (outgroup) are present in the tree. (B) Expanded look of the collapsed branch from (A). Viruses from Kazakhstan cluster together with isolates from Russia (Siberia, Urals, Caucasus) and Eastern Europe. Labels: viruses sequenced by our group are labeled with filled circles next to GISAID accession numbers. Isolates sequenced in other laboratories and downloaded from GISAID for this study are labeled with filled squares. Clusters with statistical support 99–100% are labeled B1–B5 for describing the results.

Some of the clusters defined in the HA and NA trees are actually present in phylogenetic trees for all eight segments. The Kazakhstan's representatives in the branches B2, B4, B5-2, B5-3 show the same pattern of clusterization in phylogenetic trees for the segments MP, NP, NS, PA, PB1 and PB2 (these trees are presented in the Supplementary Information).

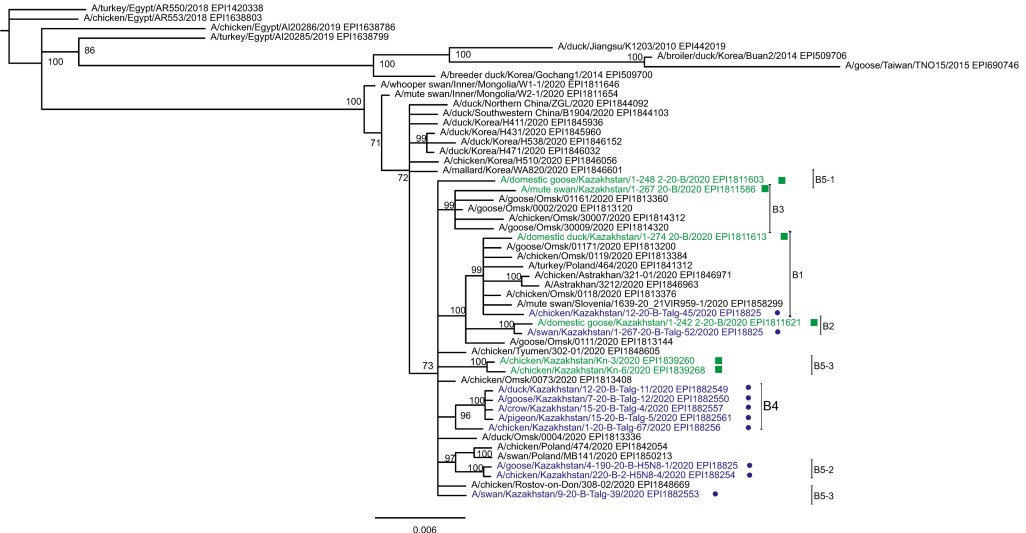

**Figure 3** **Phylogenetic tree for the neuraminidase (NA) gene.** The tree was built and labeled similarly to the tree in Fig. 2. A multiple alignment for the NA tree is presented in the Supplementary Information. NA clusters labeled B1–B4 correspond to the eponymous HA clusters in Fig. 2. Members of one HA cluster B5 in the NA tree are present as separate branches B5-1–B5-3.

Whereas the described relationships between the HPAI A/H5N8 viruses from Kazakhstan, Siberia and Europe was easy to discern, it has been difficult to discover epidemiological correlates for molecular data such as the dependence between phylogenetic clustering and sampling locations, timing, host species or case pathology. For example, the branches B1 and B2 both comprise viruses from the North and South of Kazakhstan. The branches B3, B4, B5-3 represent viruses from samples collected only in the North, and B4 (the largest cluster of Kazakhstan's viruses) represents closely related sequences in all trees. In contrast to B5-3, the branch B5-2 members are from the South. Finally, one "stray" variant in the single-tip branch B5-1 is actually from Central Kazakhstan (this sample was sequenced in the APHA laboratory). Nevertheless, the presented data confirm that the HPAI viruses in the circulation involving the territory of Kazakhstan are genetically diverse. The current knowledge on the epidemiological correlates for the Kazakhstan's HPAI viruses suffers from the real scarcity of both molecular and observational data, and the way to establish such correlations is to enhance the surveillance.

## DISCUSSION

This study purpose was to determine HPAI A/H5N8 isolates in the global contemporary circulation which show the closest relation to the viruses that caused the year-2020 outbreaks in Kazakhstan. The results reveal that the most probable route of introduction is with migratory birds.

The relation of Kazakhstan's viruses to isolates in the contemporary circulation shows that the Kazakhstan's viruses cluster with isolates from bordering regions of Russia. That is in full agreement with the epidemiological observations that the outbreaks were first

registered in the border regions. The A/H5N8 subtype is thought to have been introduced in Kazakhstan with migratory birds, highlighting the importance of continuing and strengthening the efforts to monitor influenza in migratory birds.

Viruses from Kazakhstan are grouped into several clusters with high statistical support. The majority of the clusters present in the HA tree, are also present in trees for all other genomic segments. Such grouping of the Kazakhstan's viruses suggests that the same studied outbreaks actually were a result of a simultaneous introduction of genetically distinct ancestral lines. The similarity in the grouping for different genomic segments provides an additional level of support for the described phylogenetic relationships found in this study.

The HPAI A/H5N8 viruses appear to fit for efficient conquering of new territories. To the authors' knowledge, this study is the first report describing the emergence of the A/Goose/Guangdong/96 (Gs/GD) H5 lineage in Kazakhstan.

Kazakhstan is a large country (2,725,000 sq.km, that is 61% of a size of the European Union), occupying the center of Eurasian continent and being a crossroad on ways from Eastern Europe to China in one direction, and from European Russia—Urals—Western Siberia to Central Asia in a cross direction. Whereas some parts of Kazakhstan are arid and deserted, Northern Kazakhstan is rich in lakes. In fact, a group of large lakes called Tengis—Korgalzhyn (in the national park, Korgalzhyn State Nature Reserve) have been included in the Ramsar list of wetlands of international importance. Wetland birds there are numerous and diverse as represented by 112 species, which constitute 87% of the Kazakhstan wetland avifauna. Lakes of Northern Kazakhstan are important stopping points on birds' migratory routes.

The Northern Kazakhstan natural conditions with birds crowding during mass migrations enable a rapid propagation of avian viruses. Transient coexistence in the same place of birds coming from distant habitats, and of different species, can result in mixed infections with different AIV strains and the reassortment of genome segments. By this way, a novel virus can evolve and further propagate along the migration routes.

The state veterinary surveillance system for HPAI is in effect in Kazakhstan. The routine surveillance exploits local veterinarians in settlements and administrative centers, and a central reference laboratory (NRVC) located in the capitol. Local veterinarians are obliged to monitor territories within a 10 km-radius from poultry farms and collect samples from dead birds. Also, veterinarians respond to appeals from households and farms in case of poultry die-offs, all outbreaks are registered. The collected samples are transported to NRVC for laboratory testing. The central laboratory reports HPAI outbreaks to the World Organization for Animal Health (WOAH).

However, among the most important conclusions from this study is that the surveillance system currently in place is unable to establish more detailed deterministic or statistical ties between the molecular diversity of HPAI viruses in Kazakhstan and their epidemiological/biological characteristics relevant for predicting the situation development and calculating risks. This is because of several reasons, of which worth special mentioning is the bias towards sampling domestic and industrial flocks whereas wild birds' populations remain understudied; low rates of producing molecular data; the lack of studies of the

phenotype of HPAI isolates on laboratory infection models. The latter studies are planned at the NCB.

The AIV subtype A/H5N8 was initially detected in 2010, then it caused major birds die-offs in South Korea in 2014, and since then its wide circulation has been described in the Northern Hemisphere including countries of Asia, Europe and North America (*Zhao et al., 2013*; *Jeong et al., 2014*; *Lee et al., 2014*; *Bouwstra et al., 2015*; *Hanna et al., 2015*; *Pasick et al., 2015*). However, until the outbreaks of 2020, the A/H5N8 subtype has not been found in Kazakhstan. In the Northern Kazakhstan's neighbor country Russia, A/H5N8 HPAI circulation has been recorded since 2014, however the early report described isolating of this subtype only in Russia's East, relatively far from Kazakhstan (*Marchenko et al., 2015*). The recent propagation of A/H5N8 HPAI across Eurasia was widely covered in public media, however scientific reports with the molecular characterization of Kazakhstan viruses are scarce (*Lewis et al., 2021*). The phylogenetic relationship found in this study between the viruses from Kazakhstan and Russia is expected. Russian Caspian, Russian Caucasus, Southern Ural and South-Western Siberia—all are Kazakhstan's neighbors. In fact, the border between Kazakhstan and Russia is the longest continuous land border in the world (7,591 km).

The whole Kazakhstan's territory lies under the Central Asia Migratory Flyway (also dubbed as Central Asian-Indian Flyway). This flyway connects breeding grounds along the Russian Arctic Ocean coast with overwintering places in the Indian subcontinent (*Si et al., 2009*). The outbreak began in early autumn contemporary with the beginning of birds' seasonal migration. An assumption that birds migrating from the north introduced the virus into Kazakhstan is supported by the fact that the Russian regions of Chelyabinsk (at Urals) and Omsk (at Western Siberia) had declared bird flu quarantines on August 3 and September 5, respectively, *i.e.*, before the disease came overt in Kazakhstan (*Informburo, 2020*).

With regard to mutations present in the HA protein, two rare mutations N110S and T139P reside in positions that were shown to make contacts with terminal sialic acids present on sugar chains of cell surface glycans. Thus these mutations affect the receptor recognition and may be involved in a host specificity shift. With this regard there are experimental data confirming that mutations of the sialic acid-contacting residues promote the host switch. One HA mutation in an A/H5N1 virus in a position homologous to N110S enhanced binding of the avian virus to a human-type receptor (*Su et al., 2008*). A role in shifting the viral host tropism to the human host was confirmed for the mutation T139P (*Yamada et al., 2006*).

Three mutations in NA (N46K, N84S and T295M) deserve special mentioning because these mutations remove predicted N-linked glycosylation sites and thus are assumed to diminish glycosylation in the NA stem region. There are experimental studies showing that the abrogation of the NA glycosylation enhances the virulence, increases pathology in birds and mice (*Chen et al., 2020*; *Park et al., 2017*).

However, additional experimentation would be needed to determine if the same outcomes would be observed for the Kazakhstan's A/H5N8 viruses. The whole genetic backbone of the virus including internal gene segments plays a role in whether a biological

outcome will be achieved. Indeed, further *in vitro* studies are planned to produce strains of Kazakhstan's AIVs, confirm the HPAI phenotype and study the virulence.

Being a source of economic pain for poultry producers and a threat to public health, the Gs/GD-lineage HPAI viruses is a problem worth of international attention (*Yamaji et al., 2020*). The A/H5N8 subtype is pathogenic to small mammals (*Marchenko et al., 2015*; *Marchenko et al., 2017*) and is able to infect people (*World Health Organization, 2021*). No human A/H5N8 cases have been reported in Kazakhstan before the official end of the outbreaks. As of the time this paper is prepared, Kazakhstan is officially declared to be AIV-free.

## CONCLUSIONS

The findings confirm that the A/H5N8 subtype continues its conquering epidemiological history in Central Eurasia. The surveillance and healthcare must be prepared.

## ACKNOWLEDGEMENTS

The authors acknowledge the contribution from the originating laboratories which submitted data to the GISAID database. A list of the originating laboratories for the sequences that used in this study is presented in the Supplementary Information.

### Funding
This research was funded by the Science Committee of the Ministry of Education and Science of the Republic of Kazakhstan (Grant No. AP09562122). The funders had no role in study design, data collection and analysis, decision to publish, or preparation of the manuscript.

### Grant Disclosures
The following grant information was disclosed by the authors:
Science Committee of the Ministry of Education and Science of the Republic of Kazakhstan: AP09562122.

### Competing Interests
The authors declare there are no competing interests.

### Author Contributions
- Asylulan Amirgazin, Talgat Karibayev, Maxat Berdikulov, Tamila Kozhakhmetova and Laura Syzdykova performed the experiments, prepared figures and/or tables, and approved the final draft.
- Alexandr Shevtsov conceived and designed the experiments, analyzed the data, prepared figures and/or tables, and approved the final draft.
- Yerlan Ramankulov conceived and designed the experiments, authored or reviewed drafts of the paper, and approved the final draft.
- Alexandr V. Shustov conceived and designed the experiments, analyzed the data, prepared figures and/or tables, authored or reviewed drafts of the paper, and approved the final draft.

## DNA Deposition

The following information was supplied regarding the deposition of DNA sequences:

The sequences of influenza virus genomic segments are available at the GISAID database (https://www.gisaid.org/): EPI1927648, EPI1927649, EPI1927650, EPI1882525, EPI1927651, EPI1882526, EPI1927652, EPI1927653, EPI1927654, EPI1927655, EPI1927656, EPI1882547, EPI1927657, EPI1882546, EPI1927658, EPI1927659, EPI1927660, EPI1927661, EPI1927662, EPI1882548, EPI1927663, EPI1882549, EPI1927664, EPI1927665, EPI1927666, EPI1927667, EPI1927668, EPI1882551, EPI1927669, EPI1882550, EPI1927670, EPI1927671, EPI1927694, EPI1927695, EPI1927696, EPI1882552, EPI1927697, EPI1882553, EPI1927698, EPI1927699, EPI1927700, EPI1927701, EPI1927702, EPI1882555, EPI1927703, EPI1882554, EPI1927704, EPI1927705, EPI1927706, EPI1927707, EPI1927708, EPI1882556, EPI1927709, EPI1882557, EPI1927710, EPI1927711, EPI1927712, EPI1927713, EPI1927714, EPI1882559, EPI1927715, EPI1882558, EPI1927716, EPI1927717, EPI1927718, EPI1927719, EPI1927720, EPI1882560, EPI1927721, EPI1882561, EPI1927722, EPI1927723, EPI1927724, EPI1927725, EPI1927726, EPI1882563, EPI1927727, EPI1882562, EPI1927728, EPI1927729.

## Data Availability

The GISAID accession numbers to sequences of Kazakhstan HPAI (H5N8) isolates, GISAID entities for prototypes from originating laboratories, phylogenetic trees and all multiple alignments (PB2, PB1, PA, HA, NP, NA, MP, NS) used in the study are available in the Supplemental File.

## Supplemental Information

Supplemental information for this article can be found online at http://dx.doi.org/10.7717/peerj.13038#supplemental-information.

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
