# Peer review of "Highly pathogenic avian influenza virus of the A/H5N8 subtype, clade 2.3.4.4b, caused outbreaks in Kazakhstan in 2020"

_PeerJ, doi:10.7717/peerj.13038_

## Round 0.1 · original submission · Major Revisions

The review process is now complete, and three thorough reviews from highly qualified referees are included at the bottom of this letter. Although there is considerable merit in your paper, we also identified some concerns that must be considered in your resubmission. I strongly agree with the reviewers and emphasize that the authors must dedicate themselves to answering the points raised (mainly related to the necessary robust phylogenetic analyses to be included) with precision and making all those revisions in the manuscript to be resubmitted.

·

Basic reporting

The manuscript by Amirgazin et al entitled, “Large outbreaks in wild birds and poultry in Kazakhstan during 2020 caused by the highly pathogenic avian influenza virus of the H5N8 subtype, clade 2.3.4.4b” is a study on H5N8 isolated from dead birds in Kazakhstan. The investigators performed whole genome sequencing and their analysis showed that all isolates were from the clade 2.4.4.4b and were related to other isolates from nearby geographical locations. It is the first report of clade 2.3.4.4b viruses in Kazakhstan and expands our awareness and knowledge of this lineage of avian influenza viruses. To increase readability, it would be great to have a copy-editing service, especially for parts of the Discussion. Specific comments are below:

Line 20: If I understand correctly, it should be “hit hard” instead of “hit hardly”. The phrase “hit hardly” refers to few occurrences while “hit hard” refers to many occurrences.

Line 45: The high pathogenicity avian influenza (HPAI) designation is based upon disease outcomes in gallinaceous bird species (e.g. chickens, turkeys, quail). In most cases (with exceptions), HPAI viruses only cause mild clinical disease in wetland species. It would be good to specify “flocks” as “gallinaceous flocks”.

Line 47: “sever” to “severe”

Lines 47-48: “~50% chances of death” to “high case-fatality ratio”. The reports of fatality rate are usually based upon known IAV spillover event to humans. We do not fully know how much spillover occurs as IAV is not routinely tested in the hospital/clinical setting. The use of high case-fatality ratio is thus more appropriate here.

Lines 50-52: Reassortment does occur between avian and mammalian viruses, but they are uncommon. More common reassortments are between strains from the same host. I think this sentence should also reflect the likelihood of these reassortment events.

Line 55: “ and HA (9 subtypes)” to “and NA (9 subtypes)”. Also please define HA (hemagglutinin) and NA (neuraminidase).

Line 64: “hardly hit” to “hard hit”

Line 80: Please consider using a different word in place of “epiornithia”.

Experimental design

The data and analyses presented are sound and appropriate. The methods are well-presented. Below are some specific points to address:
Line 150: “month” to “months”
Line 155: eliminate “purpose”
Line 160: Does the word “subjected” mean that the samples were also submitted, processed, analyzed in other laboratories? If so, I think a more detailed/technical word should be replace “subjected”.

Validity of the findings

1. Can you be more specific with the location each strain? Are there any epidemiological links with some of the isolates?

2. The clade/cluster assignments need to be reconsidered since some are paraphyletic, i.e., it does not include all descendants of the branch. Specifically, K8 should also include isolates from K3 and K6 since all share a common ancestor. It would thus be best to assign K8, K3, and K6 as one cluster.

3. For the discussion of mutations and associated biological outcomes, it would be best to mention that additional experimentation would be needed to determine if the same outcomes would be observed for the H5N8 Kazakhstan isolates. The genetic background plays a big role in whether a biological outcome will be achieved (also called epistasis). Indeed, it is possible to have a mutation and observe different outcomes depending on which genetic backbone is used by the virus.

Below are some specific comments:

Line 179: For influenza literature, the term “clade” has a specific definition which refers to the grouping of HA sequences according to the guidelines of the H5 evolution working group (See reference below). Therefore, it’s good practice to avoid the use of the term clade unless it refers to the HA grouping. I think it would be best to use terms such as “cluster” or “group” instead of “clades” to avoid confusion. If you perform interclade pairwise comparisons and find a >1.5% difference between isolates, then you can assign presumptive clades but eyeballing Figure 2, it appears that the Kazakhstan isolates do not satisfy these criteria.

• World Health Organization/World Organisation for Animal Health/Food and Agriculture Organization (WHO/OIE/FAO) H5N1 Evolution Working Group. 2014. Revised and updated nomenclature for highly pathogenic avian influenza A (H5N1) viruses. Influenza and Other Respiratory Viruses 8:384-388.

Figure 2 caption: “Only bootstrap values no less than 60 are shown” to “Only bootstrap values greater than or equal to 60 are shown”

Figure 3 caption:
• What is rationale behind using a bootstrap cutoff of 60 for the HA and 59 for the NA sequences?
• Are the clades/clusters indicated by K1, K2, etc. according to the HA groupings? If so, please indicate in the caption explicitly.

Additional comments

Nothing further to add

Reviewer 2 ·

Basic reporting

In their manuscript, Amirgazin et . al. present a study describing the detection and full genome sequencing of several A/H5N8 samples from Kazakhstan in 2020. A/H5N8, and other HPAI, are a global concern and the details from this paper are highly important for understanding global circulation and impact of these viruses. However, the authors could greatly improve the data included in the study and further refine the paper for overall readability and novelty.
1. Overall, the paper could use an edit for English language, grammar, syntax, and tense such as line 20, 34, 47, 48, and forward
2. Line 55: Technically there atre 18 HA subtypes and 11 NA subtypes including influenza virus family members found in bat species
3. Influenza subtypes should be reported according to generally accepted nomenclature including type as A/HxNy or A(HxNy) wherever used in the manuscript

Experimental design

4. In terms of the phylogenetic analysis, a Maximum Likelihood methodology may be much more robust and could them be coupled with a TMRCA analysis in BEAST or similar bioinformatics analysis.
5. In addition to the HA and NA, internal gene segments of avian influenza viruses are equally important for understanding reassortment, geographical spread, and even zoonotic risk. It is critical to add analysis of internal genes of these viruses as well as a tanglegram describing their relation to other global strains.
6. Individual strains are not generally considered to be individual clades, please consider revising
7. Please include a table of all mutations detected in each strain for comparison across samples described in this manuscript
8. Please ensure to include some type of indicator or text for subtype (ie: A/H5N8, A/H5N6, etc..) and GISAID or other database accession numbers in the phylogenetic trees
9. Was any further in vitro or in vivo work performed or planned to confirm pathogenicity or differences with these isolates?

Validity of the findings

No issues were determined with validity of the findings

Reviewer 3 ·

Basic reporting

Amirgazin et al., sequenced H5N8 HPAI sampled in Kazakhstan and conducted phylogenetic analysis for sequence comparison to the other H5 viruses from public database. Although the study is important in terms of an epidemiologic standpoint, data presentations are not constructive, and the structure of the manuscript needs to be significantly improved and organized for readability.

Experimental design

The methods described are not sufficient enough to cover what are shown in the results.

Validity of the findings

Throughout the manuscript, the author misused the term used in influenza virus research, and the naming of the clade classification of GsGD H5 lineage virus does not follow the recent WHO guidance.

Additional comments

line by line comments

Line 43: The two references used to demonstrate the natural reservoir of AIV should be replaced with other references. They are not comprehensive, and one of them is irrelevant.
Line 44: Poultry also can be infected with IAVs, but not all IAV. Generally, IAVs that can infect and productively circulate in wild birds and poultry are called Avian influenza virus (AIV). It is also defined by the tripartite international organization (WHO, FAO, and OIE). Throughout the manuscript, the authors should have used the term, Avian Influenza virus (AIV), not IAV.
Line 50: Again, IAV represents all influenza type A viruses.
Line 71-72: If the outbreaks are still ongoing and declared HAIV free, state it more clearly as of the time this manuscript has been written.
Line 75: What the recent prototype sequences mean is unheard of. If you used previously isolated sequences from the public database such as GISAID or IRD, make it more transparent.
Line 76-77: What is more important thing than that specific subtype (H5N8) has never been detected in Kazakhstan is that the A/Goose/Guangdong/96 (Gs/GD) H5 lineage has ever been introduced to Kazahstan. Has ever Gs/Gd H5 lineage virus been detected in Kazahstan before?
Line 77-78: Determination of basic amino acid in the HA cleavage site is the result of this study.
Line 84-85: What were the species of dead birds? How many dead birds were sampled? Was there any clinical complications found at the time of dead birds found?
Line 119: Does the identifier present the accession number given by Genbank or GISAID? Then, add strain name along with the number.
Line 125, 128: Using the sequences deposited during time doesn’t make sense. It should be isolated.
Line 143-145: Could you provide an approximate number of wild birds affected during the die-off event?
Line 149: What is the calculation of the number of affected poultry? Were there any measures taken, such as culling poultry from the affected farm? In that case, does the 2 million include the culling?
Line 151: Eleven out of fourteen?
Line 157-158: Since 1976 out of 2212 carcasses tested positive, what was the criteria that the ten viruses were picked up for whole genome sequencing?
Line 157: because we don’t know if the viruses sampled were viable by only doing the RT-PCR and the viability was not confirmed by egg inoculation, the term ‘isolate’ should’ve been used. This term is found throughout the manuscript.
Line 159-164: The other viruses sequenced elsewhere is not part of this study. Other sequences used for phylogenetic analysis should be in the materials and methods.
Line 167-168: Looking at the phylogenetic grouping, the H5 subclade classification used in this study referred to some previous papers published when early clade 2.3.4.4 viruses emerged (clade 2.3.4.4 group A – clade 2.3.4.4 group D). However, this classification was updated in detail as the viruses diverged. Please use the latest classification (clade 2.3.4.4a – clade 2.3.4.4h) as the factsheet (https://apps.who.int/iris/handle/10665/336259)
Line 174: How many nucleotide differences were found in NA?
Line 175: Why are there 16 NA sequences? Only ten viruses were sequenced in this study.
Line 179-190: For the Gs/Gd H5 lineage, the clade classification means the sub-lineages of H5 HA genes. The K1 – K8 naming should be cluster. In addition, looking at the phylogenetic HA and NA trees some clusters does not supported by a bootstrap value (generally bootstrap value >70 is acceptable for clustering).

---

## Round 0.2 · Minor Revisions

The authors addressed the main concerns of the reviewers and, I feel the manuscript is much improved by the modifications. However, the revised manuscript still deserves attention.

Please, provide point-to-point responses according to the comments made the Reviewer #1 in the new version of your manuscript.
I also identified some concerns that must be considered in your resubmission. (a) Abstract: add a sentence in the conclusion topic regarding the impact of the findings in the context of public health. (b) Discussion: briefly present the novelty of the article based on the data in a new first paragraph. Discuss in more depth the importance of the findings for public health. (c) Improve the quality of Figure 1: make the map cleaner to highlight the studied locations; do not show the regions in Russia (no importance in the context); add the other bordering countries for better geo-referencing. (d) Tables 1 and 3: replace the commas to periods for the percentages shown.

·

Basic reporting

The manuscript by Amirgazin et al entitled, “Highly pathogenic avian influenza virus of the A/H5N8 subtype, clade 2.3.4.4b, caused outbreaks in Kazakhstan in 2020” presents the detection of clade 2.3.4.4b H5N8 high pathogenicity avian influenza viruses in Kazakhstan from various avian species. Phylogenetic analysis of the sequences demonstrate that the viruses sequenced were related to viruses isolated in neighboring countries. This manuscript is a major improvement over the prior version in methods used for phylogenetics, content, and writing style.

Experimental design

The use of Bayesian phylogenetics is a much appropriate method to analyze avian influenza sequences. The description of the methods used are sufficiently detailed to be able to reproduce the results.

Validity of the findings

The assignments of putative phylogenetic groups according to the HA tree are more appropriate although it is unclear what is the epidemiological significance of these groupings. Do the samples in each HA phylogenetic grouping share a common geographic location and/or species from which the samples were taken from? Is the linkage temporal? I would suggest addressing the epidemiological significance of the B1-5 groups in the discussion section.

Additional comments

No additional comments

---

## Round 0.3 · Minor Revisions

The authors responded to the raised questions and made the necessary changes to the manuscript. Before accepting the paper, delete the second paragraph (lines 295-310) from the Discussion Section.

---

## Round 0.4 · accepted · Accept

The authors made the required change to the manuscript.